# Analysis of Mechanical Properties Related to Fiber Length of Closed-Loop-Recycled Offcuts of a Thermoplastic Fiber Composites (Organo Sheets)

**DOI:** 10.3390/ma15113872

**Published:** 2022-05-29

**Authors:** Sabine Hummel, Katharina Obermeier, Katja Zier, Sandra Krommes, Michael Schemme, Peter Karlinger

**Affiliations:** 1Department of Plastics Technology, Faculty of Engineering Sciences, Technical University of Applied Sciences Rosenheim, Hochschulstraße 1, 83024 Rosenheim, Germany; sabine.hummel@th-rosenheim.de (S.H.); peter.karlinger@th-rosenheim.de (P.K.); 2Department of Sustainable Engineering & Management, Faculty of Management and Engineering, Technical University of Applied Sciences Rosenheim, Hochschulstraße 1, 83024 Rosenheim, Germany; katharina.obermeier@th-rosenheim.de (K.O.); zk.zier@gmail.com (K.Z.); sandra.krommes@th-rosenheim.de (S.K.)

**Keywords:** recycling, glass fibers, polypropylene, organo sheet, closed loop, hybrid process, injection molding, life cycle assessment, economic efficiency

## Abstract

Increasing demand for energy-efficient means of transport has steadily intensified the trend towards lightweight components. Thermoplastic glass fiber composites (organo sheets) play a major role in the production of functional automotive components. Organo sheets are cut, shaped and functionalized by injection molding to produce hybrid components, such as those used in car door modules. The cutting process produces a considerable amount of production waste, which has thus far been thermally recycled. This study develops a closed mechanical recycling process and analyzes the different steps of the process. The offcuts were shredded using two shredding methods and implemented directly in the injection-molding process. Using tensile tests and impact bending tests, the material properties of the recycled materials were compared with the virgin material. In addition, fiber length degradation via the injection-molding process and the influence of the waterjet-cutting process on the mechanical properties are investigated. Recycled offcuts are both comparable to new material in terms of mechanical properties and usability, and are also economically and ecologically advantageous. Recycling polypropylene waste with glass fiber reinforcement in a closed loop is an effective way to reduce industrial waste in a sustainable and economical production process.

## 1. Introduction

Thermoplastic glass fiber composites are light and have excellent mechanical properties and are therefore commonly used to form automotive components [1,2,3]. In particular, the manufacturing of hybrid components that combine the advantages of organo sheets with the possibility of functionalization by using injection molding has gained significance over the last decade [2]. The carrier (Figure 1) was made of compressed woven glass fibers embedded in a thermoplastic matrix (organo sheet). This flat semi-preg has high strength and stiffness values combined with low density, and can be used in the thermal-forming process. Injection-molded elements made from long fiber thermoplastics granulate (LFT) can be produced in high quantities and easily functionalized. The injection-molding process combines the thermal forming of the organo sheet with the subsequent injection and overmolding of the LFT material. The result is a product that has the benefits of both processes. Continuously reinforced, pressed semi-preg products offer good stiffness. The material is used in injection molding to add functional elements such as functional carriers, ribs or edge structures made from LFT [1,4,5,6]. These hybrid components can be produced in large quantities at low cost, which qualifies them as a substitute for components made from steel and other composite materials [2,7]. The components are produced in a nesting process, in which the organo sheets are cut into the desired geometry using a load-adapted waterjet cutter. The nesting process, however precise, always produces waste offcut [8].

There are different ways to treat this post-industrial waste. In the past, material waste with a high inorganic content was simply deposited in landfills. Due to European legislation, materials with an inorganic content above the threshold of 5 percent by weight may no longer be discarded in landfills. First, the material must be pretreated to reduce the inorganic content, which ejects greenhouse gases when scrapped. Incineration or co-incineration for cement production is the current means of waste treatment [10,11].

During incineration, the high calorific value of the waste is used to recover thermal energy, and the glass fibers end up partly in the slag [12]. The latter is then deposited in a landfill site at high cost [11]. However, some of the fiber material can cause problems during incineration and exhaust gas treatment. Therefore, co-incineration for cement production is the preferred solution for waste treatment. The material waste replaces the fossil energy needed to burn the lime, and the glass fiber is used as a filler in cement, which can be used as an ingredient in concrete [2,11,13].

According to the waste hierarchy, recycling of the material must be assessed before any other recovery. Possible processes are feedstock and material recycling. Feedstock recycling includes processes such as solvolysis and pyrolysis. Solvolysis is a chemical decomposition process, in which the polymer matrix is dissolved resp. depolymerized by chemicals [11,14]. Pyrolysis, on the other hand, removes the polymer matrix by combustion, microwave-assisted heating or fluidized bed extraction [11,13,14,15]. In both processes, the end products can be further processed. Feedstock recycling processes are currently not industrially relevant, either because there are only pilot plants, the process costs are too high or the recycled materials have high impurities which requires a reprocessing of the recycled products [11,13,15,16].

Mechanical or material recycling of the offcuts is a promising recycling option. Depending on whether the material offcuts are recycled in the same process or in a different process, it is defined as closed-loop or open-loop recycling. In material recycling, the waste is first shredded, cut or ground. The recovered material can then be reused [2,11,13,14,17,18,19,20]. To demonstrate the mechanical recyclability of composites with thermoplastics, Otheguy et al. 2009 recycled the hull of a boat made from PP-GF laminate with a balsa wood core [21]. Luinge and Warnet 2020 implemented shredded thermoplastic laminates as the middle layer in a new fiber composite laminate [20]. A similar application was studied by Kiss et al. 2020. They compared the mechanical properties of sandwich laminates filled with shredded thermoplastic laminates with those of panel-molded compounds made of shredded thermoplastic laminates. The sandwich construction achieved better results due to the continuously reinforced top and bottom layers. Kiss et al. also tested the shredded material in injection molding and found a decrease in impact strength due to fiber abrasion [19]. Further research has dealt with recycled thermoplastics composites and has compared two types of recycling materials, shredded laminates and cut semi-pregs with novel LFT pellets by processing in a low-shear mixer and subsequent compression molding [17,18]. The characteristics (fiber orientation, percolation, fiber attrition, etc.) of the recycled material were comparable to those of virgin glass-mat thermoplastics (GMT) products [17]. Bernasconi et al. 2007 investigated the processing in injection molding, e.g., the admixture of short glass-fiber-reinforced polyamide 6.6 recycling material compared to virgin material with different portions of 0%, 25%, 50% and 100%. They specialized in the fatigue strength of the material mixtures [22]. Evens et al. 2019 investigated the properties of components made of LFT granulates that have undergone up to ten recycling cycles. For the polypropylene with glass fiber, they found that the mechanical properties decrease with each recycling step due to fiber length degradation [23].

This study incorporates the subsequent results as part of a research project that deals with closed-loop recycling, which investigates the recycling of offcuts from thermoplastic fiber composites with continuous reinforcement. The closed-loop recycling of a series-produced hybrid door module was used as an example. Here, organo sheet offcuts were fed directly into production in order to replace primary material for the injection molding of functional elements of the component. The continuous reinforcement of the offcuts promises high potential by achieving the longest fibers possible in manufactured components. This promotes high quality and material recycling where long-fiber glass material can be replaced with consistent mechanical properties. Currently, only short glass-fiber-recycled materials are available on secondary material markets; however, these do not fulfill the mechanical-property profile of the functional elements.

Two comminution methods for the production of recycled materials with long fibers, and their processability in the injection-molding process, were dealt with in this study. First, the mechanical properties of the recycled materials were analyzed and compared with reference materials available on the market. In the second step, optical and thermal analyses were carried out in order to be able to compare the properties in terms of fiber lengths and fiber content of the recycled material with the primary material. A correlation between the fiber analysis and the mechanical properties of the materials can be established with these two steps. In addition, the ecological and economic impact of closed-loop recycling was analyzed via a life cycle and profitability assessment. Closed-loop recycling of the offcuts can be recommended from an ecological and economic point of view, and the mechanical properties are also sufficient due to achievable fiber lengths.

## 2. Materials and Methods

A suitable strategy was developed for the recycling process of organo sheets, which was achieved by processing the nesting offcuts into recycling flakes by means of crushing processes. Two of these crushing processes were specifically compared. The crushed flakes were then introduced into the injection-molding process using a direct-injection-molding method, which serves as the functionalization of the hybrid component “hybrid door module”. The primary long-glass-fiber granulate was thus replaced. To ensure a glass fiber weight percentage of 30%, a base polymer and an adhesion promoter were added to the recycling organo sheet flakes.

### 2.1. Materials

The raw material used in this study is the organo sheet semi-preg (TEPEX Dynalite 104ms-RG601(1)/64mf% black Type S) from Bond-Laminates, consisting of 64 weight percentage of woven glass fibers and 36 wt.-% of the matrix material polypropylene (PP). Furthermore, a primary long-fiber injection material made of 30 wt.-% glass fibers and polypropylene, abbreviated as: LGF30 (GD301FE), which served as a reference for the experimental results, was provided by Borealis. A regranulated material from the company Wipag was used as the second reference material. This regranulated material was produced from organo sheet offcut. The regranulated material is a short glass fiber granulate with a 30 wt.-% glass fiber content based on polypropylene, abbreviated as: RG. By shredding the offcut of the organo sheet, the resulting material has a higher amount of glass fibers; when comparing it with the reference, it must be diluted with neat polypropylene (HJ120UB, Borealis). In addition, a coupling agent has to be used to achieve a better adhesion between the matrix and the fibers (WG068AE from Borealis).

### 2.2. Comminution Methods

Principles of cutting, punching or grinding are used for various comminution tasks in the industry. Examples of this can be seen in the production of gravel, crushed stone, common building materials, cement or coal. Such processes are used in the recycling of end-of-life products, e.g., clothes, tires, glass. Shredding is also necessary in the food industry and for the production of animal feed. Shaft shredders, hammer and impact mills, rotor shears, cutting mills or punches are often used. In plastics processing, thermoplastic production rejects or start-up materials can be grinded during batch changes and subsequently melted down.

In order to be able to process the organo sheet offcut in the injection-molding process, they have to be shredded to flakes. The challenge here is to get the recycling flakes as large as possible and yet as small as necessary. The size of the recycling flakes can adversely affect dosage and large flakes can clog the draw screw during processing. Another criterion for recycling flake dosage is the condition of the cut edges. Cutting edges should be as clean as possible and without protruding glass fiber to prevent bridging. Two comminution methods were chosen and compared to produce the following results. Figure 2 shows an example of the composition of the recycling flakes produced by the two crushing processes.The first process is a customized tool system. Here, the flakes are punched into defined squares by means of a punch die. The size should ideally be 12 mm × 15 mm. As shown in Figure 2a, the edges have been defined and no fibers should be peeking out. The result is so-called punched flakes. These flakes can be processed very adequately.

The second process is a 4-shaft shredder. Here, the flakes are shredded between counter-rotating shafts that are equipped with cutters, then specially perforated sieves are used to select the size. A perforated 20 mm sieve was used for this study. Shredded flakes were created (Figure 2b) using this process. The edges of shredded flakes are well defined with only a few exposed fibers, which could lead to bridging in the dosing process. The shredded flakes have an inhomogeneous size distribution because of fractionation with a sieve.

### 2.3. Further Processing Methods

The injection-molding tests were carried out on a KraussMaffei injection-molding-machine (type: GVX 451 (DCIM), KraussMaffei Group GmbH, München, Germany) in the technical center of the Rosenheim University of Applied Sciences. The material combinations are fed to the plasticizing cylinder with a gravimetric-dosing system from Motan and injected to form multitest specimens. These series of tests were carried out with a three-zone screw (standard injection-molding process). The focus was on a process that simulates series production; therefore, all tests were carried out with conventional injection-molding processes.

The organo sheets have a glass fiber weight content of 64 wt.-%. For the functionalization in subsequent production components, a glass fiber weight percentage of 30% was to be achieved. For this purpose, a neat PP and an adhesion promoter were added to the recycling flakes to achieve the desired fiber content (Figure 3). The recycling flakes, as well as the other components mentioned above, were added directly in the injection-molding process.

The recycling flakes as well as the other components mentioned were added directly in the injection-molding process. Figure 4 clearly shows the examinations of the material combinations.

For material testing, multifunctional test specimens were produced and the mechanical properties were tested in accordance with common standards. In addition, initial fiber length analyses were carried out. The testing specimens were tested with the tensile test method according to DIN ISO 527-2 (Zwick/Roell type Z020/20kN, ZwickRoell GmbH & Co. KG, Ulm, Germany). Furthermore, an impact bending test (ZwickRoell-HIT50P/5J, ZwickRoell GmbH & Co. KG, Ulm, Germany) according to DIN ISO 179 was carried out. For the determination of the glass fiber content in the component, the samples were additionally ash-tested according to DIN EN ISO 1172.

The fiber length analysis was determined with the FibreShape CROSS system (from IST AG company, Ebnat-Kappel, Switzerland). For this purpose, the samples were first pyrolyzed and analyzed with FibreShape CROSS. Presentation of the measurement results was carried out according to ISO 9276. The fiber length degradation during the injection-molding process was analyzed, and samples were taken for analysis at three separate process points per test series.

Non-destructive images were taken with a computer tomograph (Werth Messtechnik GmbH, Gießen, Germany) for the optical evaluation. In order to achieve the highest possible resolution, and to get as close as possible to the tube, tension rods of the multitest specimens component were used. A section tomography was taken with an aluminum filter with a resolution of 4.5 µm voxel size.

#### 2.3.1. Mass Flow

The list of mass flows based on the example of the production of a hybrid door module was used as a basis for the economic and ecological evaluation of the closed-loop process developed in the project, as well as for the evaluation of the proportion of recycled material that can be technically used in the example of the door carrier module carrier.

A mass flow calculation (related to one production year) was carried out. Taking into account an assumed reject rate of 5 wt.-% during the comminution of the organo sheet offcuts into regrind, a substitution of 32 wt.-% of the total gating material used by the recycled material (regrind) could be realized. A (neat) PP and a masterbatch (adhesion promoter) were added to the regrind (glass fiber weight content of 64%) to obtain a glass fiber weight content of 30% in the recycled material. The remaining 68 wt.-% of the total feedstock material used must be primary material (LGF30). The process and mass flows are shown schematically in Figure 3.

Substitution potential of primary material, as well as a flat-rate scrap of 5 wt.-% of the regrind, was also taken into account in the following economic and ecological evaluation. During the technical implementation of the injection-molding testing at the Technical University of Applied Sciences Rosenheim, series of tests were carried out with a proportion of 33 wt.-% recycled material. Multifunctional test specimens were used to perform extensive mechanical testing on recycled materials for these test series.

In the case of the door carrier module, a maximum attainable recycled-material content of 35 wt.-% resulted from the mass flow calculation (33 wt.-% is used in the tests). Other components may also have higher recycled content, which is why a recycled content of 67% or a complete substitution of the primary material (100% recycled content) were also used as comparative values in the test series. These test series were conducted and compared with both comminution methods.

#### 2.3.2. Influence Cutting Sand

The organo sheets were cut using an abrasive waterjet-cutting process designed for the hybrid component production process. However, this resulted in impurities of organo sheet offcuts. An additional cleaning process for the offcuts would not be logistically or economically advantageous. Therefore, additional test series were carried out to investigate the influence of contamination on the mechanical properties of the recycled materials. These tests were performed exclusively with punched flakes. Images were taken with a Tescan scanning electron microscope (SEM) (Tescan type: MIRA3, TESCAN GmbH, Dortmund, Germany) for the optical assessment of the cutting sand and the contamination after waterjet cutting.

## 3. Results and Discussion

### 3.1. Mechanical Properties—Mixed Materials

Figure 5 shows the tensile strength and impact strength of various material mixtures. A total of 40 test specimens of each material mixture combination were tested and the mean value was plotted with the corresponding standard deviation (*n* = 40). All three mixing variants (100 wt.-%, 67 wt.-% and 33 wt.-% recycled content) as well as both comminution methods (regrind) and virgin LGF30 are situated between the lower reference (RG) and the upper reference (LGF30) in terms of tensile strength and impact strength. The comminution method appears to have no significant influence on the mechanical properties. Both the shredded flakes and the punched flakes are well suited for these processes. De Bruijn et al. 2017 had come to the same conclusion by testing different input forms of recycled laminates [18]. Direct processing of the shredded organo sheet offcuts with conventional injection-molding machines are meaningful from a technical point of view. It can be clearly seen that direct processing of crushed material has better mechanical characteristics than regranulated material (RG). The amount of primary material added to the recycled material has no significant impact. Recycling of organo sheet cutoff is still worthwhile even without taking into account the quantity of the actual mass flow.

### 3.2. Fiber Length Degradation

The length of the recycled glass fibers is decisive for the strength of the hybrid components made from organo sheet offcuts. This connection was demonstrated by Thomason et al. 1997 with the investigation on fiber length and the concentration of glass-fiber-reinforced materials [24]. The FibreShape CROSS system allows fiber lengths to be analyzed promptly and the closed-loop recycling to be monitored in several process steps. Firstly, a starting material, such as plastic granulate or recycling flakes, is taken as a sample. Then, additional samples are drawn from the strand (between the injection-molding nozzle and the mold) and from the finished component. Doing so establishes a synergy between fiber length degradation and mechanical properties in the component and recycled material. By comparing the length measurements, fiber length reduction is quantified and controlled, and allows the observation of a distinct reduction in fiber lengths due to the injection-molding process. This observation has already been made by several researchers [22,23,25].

Figure 6 shows the fiber length degradation of the recycled materials with the shredded flakes and the punched flakes. The glass fiber length degradation of these two comminution methods is compared with the upper (LGF30) and the lower (RG) reference material. A large scattering of fiber lengths can be seen in the base material of the recycled flakes. The mean value of the fiber lengths of the punched flakes is just under 12 mm, which can be explained by the defined punching method employed. The shredded flakes, on the other hand, were shredded in an uncontrolled manner by cutting shafts, and explains why the position and orientation of the fibers is not homogeneous in this case. After shredding, shorter fibers may well have been present in the starting material. Therefore, the mean value of the fiber lengths of the shredded flakes only shows an average fiber length of approx. 5 mm. Fiber length reduction is the same regardless of the nature of the recycling flakes. As can be observed in the technical parameters in Figure 6, the fiber lengths in the molded component (after the injection-molding process) are also between the lower (RG) and the upper (LGF30) references. Here, a synergy with the mechanical properties can be established.

### 3.3. Influence of Impurities from Separation Processes

During the waterjet-cutting process, organo sheet remnants were contaminated by sand and chips resulting from a mixture of polymer and glass fibers. This contamination, including the original cutting sand, was meticulously examined under a scanning electron microscope (SEM) (Figure 7a cutting sand and Figure 7b contamination).

The original cutting sand (neat waterjet cutting sand (a)) has sharp edges. This explains the influence on the mechanical properties when processing contaminated recycling flakes (Figure 8). One explanation is that the glass fibers are broken by contamination during melting in the plasticizing process, which occurs by the shearing of the feed screw and through the geometry of the existing cutting sand. Figure 7b clearly shows broken fibers; however, this picture was taken before the recycling flakes were processed.

Figure 8 demonstrates the mechanical properties (tensile strength and impact strength) of cleaned and contaminated punched flakes compared to the lower (RG) and upper (LGF30) references. As already described, it can be observed in Figure 5 that the cleaned recycling flakes have better mechanical properties. The series of tests shown in Figure 8 were carried out specifically with punched flakes and demonstrate that cleaning organo sheets of residue increases the benefits of the mechanical properties.

### 3.4. Visual Assessment

Additional recordings were made using a computer tomograph from Werth Messtechnik. Figure 9 compares the reference materials (RG and LGF30) and the blended variants 100 wt.-%, 67 wt.-% and 33 wt.-% of the shredded flakes. All images were scanned and evaluated with the same settings and resolution to establish a direct comparison. The measurements obtained by computer tomography confirm the results of the mechanical tests. There is no difference between the material mixtures, which means that fiber orientation and fiber lengths are almost identical. Likewise, a significant difference in fiber lengths can be seen in the images of the reference materials (Figure 9a,b) This finding was previously described in Section 3.2, and thus establishes yet another correlation to the mechanical properties (Section 3.1). In image (a) LGF30 below, slightly longer fibers can be seen when compared to images (c) 100%, (d) 67% and (e) 33% of the mixed materials. This also confirms the results of the fiber analysis in Section 3.2 (Figure 5) and the mechanical properties in Section 3.1 (Figure 5).

### 3.5. Ecological and Economic Assessment

Evaluating the ecological and economic benefits of a process are essential before practical implementation can begin. This study therefore took these two additional aspects into account, while evaluating multiple impacts of a closed-loop recycling process of organo sheet offcut, alongside the technical results. Based on the cradle-to-gate system boundary, the environmental impacts of material recycling were compared with the co-incineration of the offcuts in a cement plant, and the processing costs of both recycling methods were evaluated (Figure 10).

#### 3.5.1. Profitability Assessment

Based on recent prices for virgin materials, recycling of industrial waste and components has not been economically viable to date. However, increasingly rising prices and decreasing supply of raw material have been predicted, which absolutely promote closing the life cycle loop for the future. Furthermore, environmental legislation and social awareness have urged companies to recycle materials and use recycled materials in their products, e.g., the German “Kreislaufwirtschaftsgesetz” (KrWG). Based on the norm VDI 2243, material cycle suitability is calculated to rate the economic effect of the recycling of the offcut material. In this norm, material cycle suitability is defined as the ratio of virgin material and disposal costs (co-incineration in a cement plant, for example) to process costs of material recycling: a value equal or higher than 1 indicates that material recycling is economically viable. The process costs (closed loop) include the invest of a shredding facility, the transportation of materials, operating costs and material mixing. In contrast, for the disposal by third parties (standard), the price for virgin material and for co-incineration add up. Due to price volatility, which has varied between EUR 1.50 and EUR 2.50 per kg in recent years, material cycle suitability is plotted as a function of the virgin material prices (Figure 11) [26].

Even the relatively modest utilization of shredding facilities, which means processing at least 10% of the organo sheet offcut, would already achieve a material cycle suitability comparable to the virgin material price of EUR 0.50 per kg. Rising prices for virgin materials make material cycle suitability even more commercially attractive than before.

#### 3.5.2. Life Cycle Assessment

A life cycle assessment (LCA) using the cradle-to-gate system was made according to the international norm DIN EN ISO 14040/14044. The current, standard situation covers processing steps required for raw material extraction, production of semi-finished products, hybrid component production using injection molding and the co-incineration of offcut in a cement plant. Co-incineration replaces fossil fuels, and for this a credit is granted in the LCA on the basis of calorific value. In the closed-loop system, instead of co-incineration, shredding and pretreatment of the offcut material have been (Figure 9) investigated. The functional unit of 1000 units was used for this purpose to quantify input and output flows in the life cycle inventory. For the life cycle impact assessment, the method CML2001 was applied to compare the environmental impact of both systems based on the categories of abiotic resource depletion (ADP) elementary and fossil, the eutrophication potential (EP), human toxicity potential (HTP), ozone depletion potential (ODP), global warming potential (GWP) and the acidification potential (AP). Figure 12 shows the differences between the standard system (co-incineration of the offcut in a cement plant) and the closed-loop recycling process. Compared to the current standard of co-incineration of offcut materials, the closed-loop recycling process achieves a reduction in the environmental impact in all listed categories. In particular, a decrease of 13.3% could be achieved in the abiotic resource depletion (ADP), and the global warming potential (GWP100) decreased by 6,6%. Thus, the reduced environmental effects from the substitution of virgin material exceeded the substitution of fossil energy in the cement production process. Closed-loop recycling is therefore also worthwhile from an environmental point of view.

## 4. Conclusions

In this article, a mechanical recycling of organo sheet offcut was investigated with two different comminution methods: shredding and punching. Regarding tensile and impact strength as well as fiber length, both methods provided material with good properties fit for injection-molding processes to produce a hybrid component. The offcuts were made of continuous fiber-reinforced thermoplastics and allowed for the fiber length required for functionalization. As a result, this recycled material can technically compete with virgin material. Furthermore, the profitability analysis revealed a good return on investment while the life cycle assessment showed a reduction in environmental impact through the recycling of the offcut.

In conclusion, the circulation of the material produced by the cutting process of the organo sheets is a good technical solution for industrial waste. Virgin material can be substituted with recycled which reduces resources, costs and the impact on the environment.

Further investigations could focus on the production of the longest fibers possible in the continuous fiber-reinforced thermoplastics. The optimization of feed-screw geometry and machine configuration could deliver even better mechanical results than were achieved in this study. Recycling valuable and high-quality materials such as organo sheets would become even more advantageous.

Further research should address the scaling of the closed-loop recycling process and reliable supply chains for recycled materials which are required to bring these measures into a series production process.

## Figures and Tables

**Figure 1 materials-15-03872-f001:**
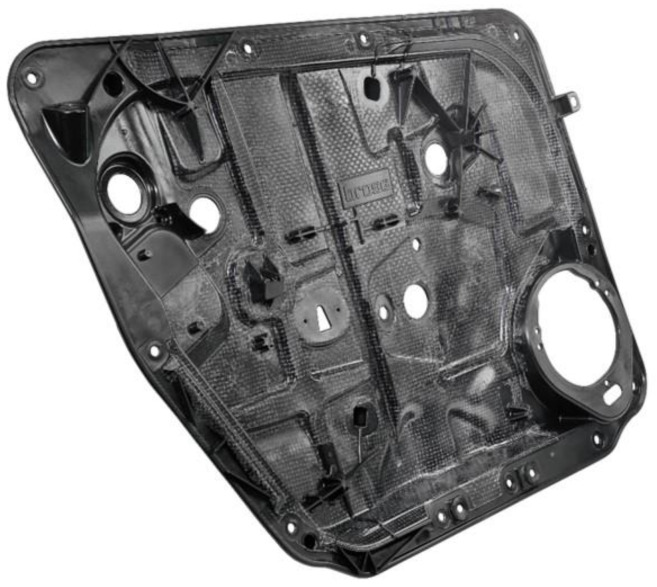
Typical door module carrier hybrid-component made of polypropylene with woven-glass-fiber-reinforced organo sheet and injected with structures (ribs, functional elements, edges, etc.) Reprinted/adapted with permission from Ref. [9]. 2021, ElringKlinger.

**Figure 2 materials-15-03872-f002:**
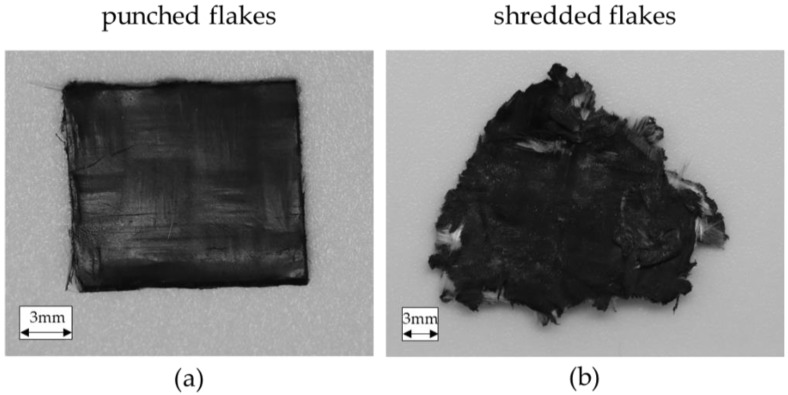
Comparison of the recycling flakes from two crushing processes, (**a**) punched flakes and (**b**) shredded flakes.

**Figure 3 materials-15-03872-f003:**
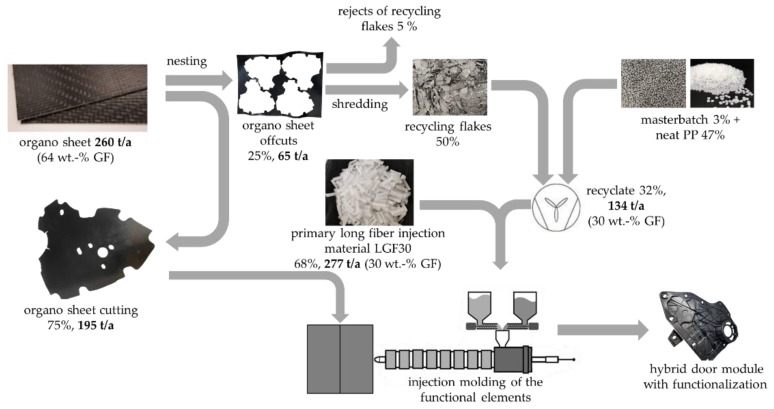
Recycling process and mass flows in the production of a hybrid door module considering closed-loop recycling of organo sheet offcuts.

**Figure 4 materials-15-03872-f004:**
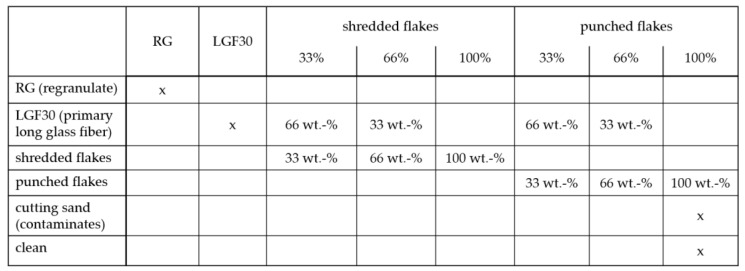
Summary and overview of the material variants investigated.

**Figure 5 materials-15-03872-f005:**
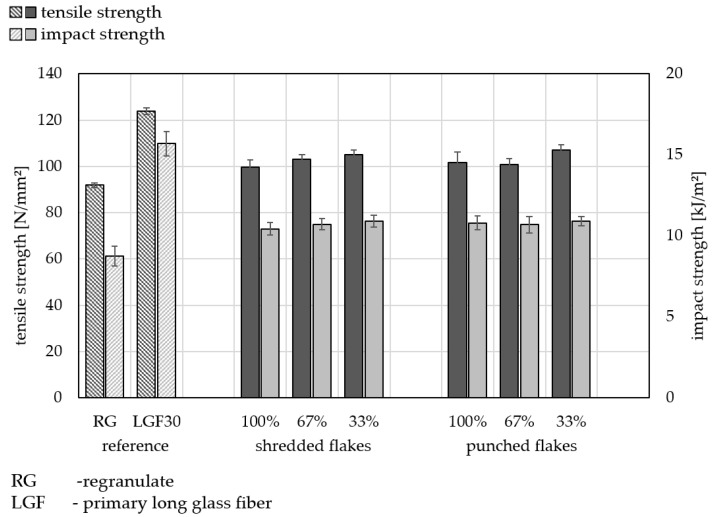
Comparison of the mechanical properties of the comminution methods (shredded flakes and punched flakes), the mixed materials (100 wt.-%, 67 wt.-% + 33 wt.-% LGF30 and 33 wt.-% + 67 wt.-% LGF30 recycled content) of the organo sheet offcuts based on the door carrier module and the references virgin LGF30 and the regranulated offcuts (*n* = 40).

**Figure 6 materials-15-03872-f006:**
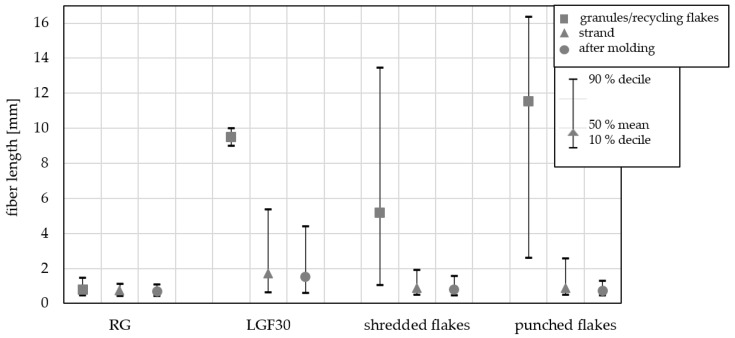
Comparison of fiber length degradation during injection molding between the two comminution methods (shredded flakes and punched flakes) and the reference materials (RG and LGF30); shown are the deciles 90% and 10% as well as the mean value with 50% of the fiber lengths per sample.

**Figure 7 materials-15-03872-f007:**
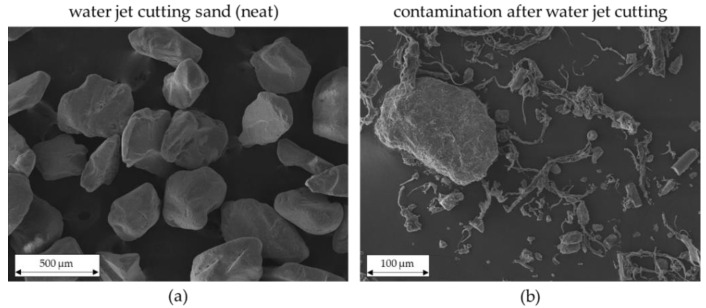
SEM images (**a**) neat waterjet cutting sand; (**b**) contamination found in the flakes after waterjet cutting.

**Figure 8 materials-15-03872-f008:**
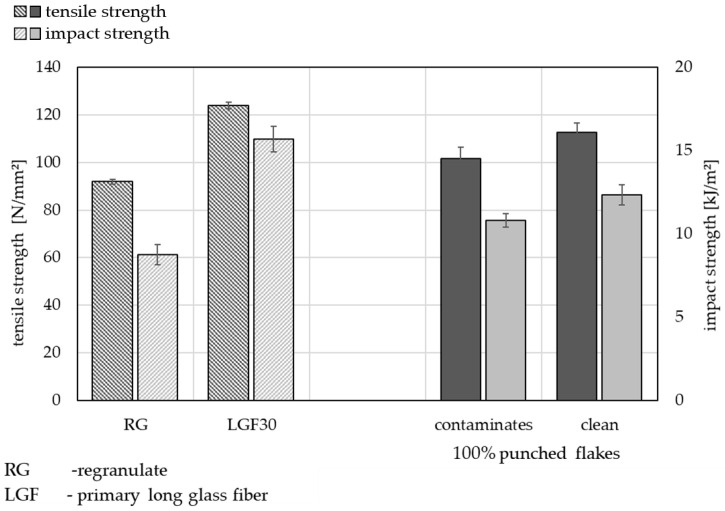
Influence of the impurity from the waterjet-cutting process on the mechanical properties (tensile strength and impact strength) in the processed materials (with punched flakes only) in comparison with the references RG and LGF30.

**Figure 9 materials-15-03872-f009:**
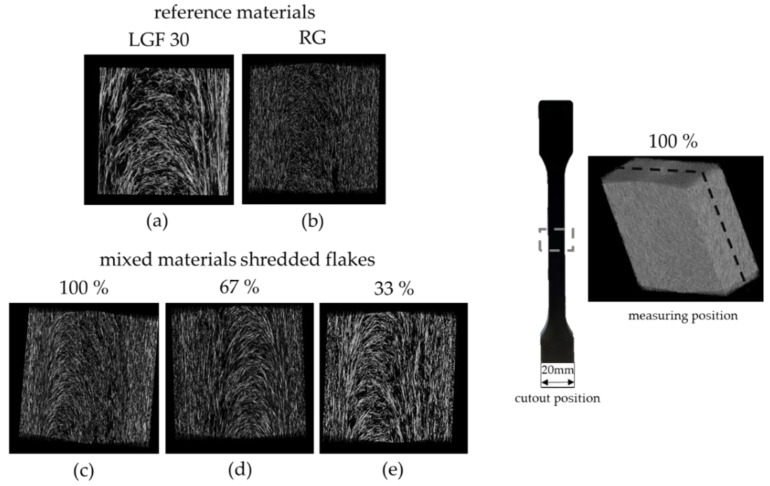
CT images comparing the fiber lengths and fiber orientation of the reference materials ((**a**) LGF30 and (**b**) RG) and the mixed materials with the shredded flakes ((**c**) 100 wt.-%, (**d**) 67 wt.-%, and (**e**) 33 wt.-% of recycled material); with a resolution size of 4.5 µm voxel.

**Figure 10 materials-15-03872-f010:**
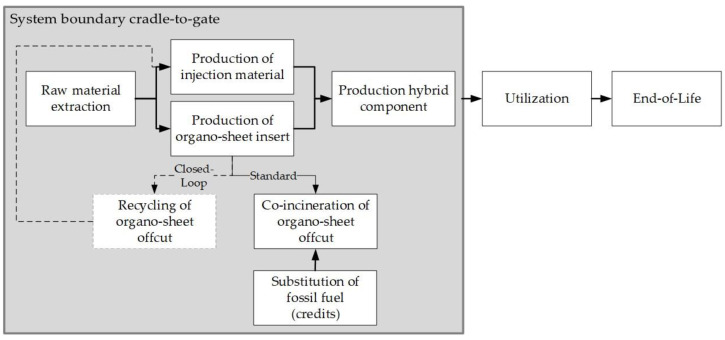
System boundary cradle-to-gate of the production of hybrid component for profitability and life cycle assessment of the considered strategies for organo sheet offcut.

**Figure 11 materials-15-03872-f011:**
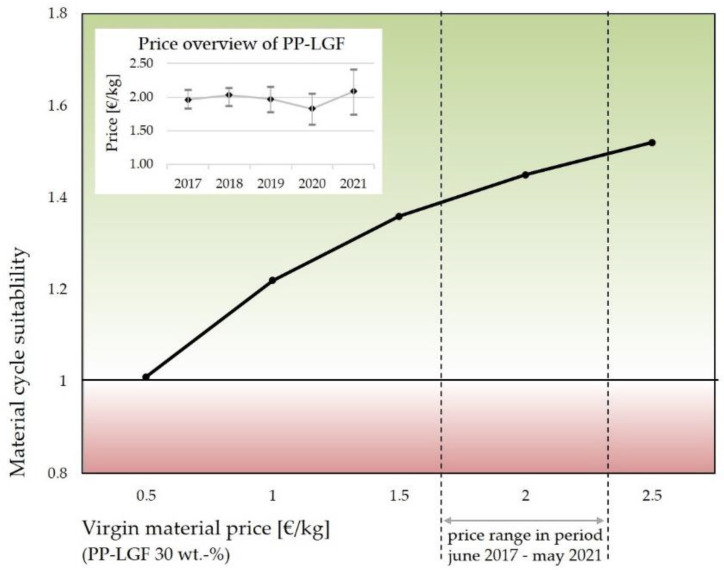
Material cycle suitability for material recycling of organo sheet offcut, which depends on the price of virgin material for PP-LGF (30 wt.-%).

**Figure 12 materials-15-03872-f012:**
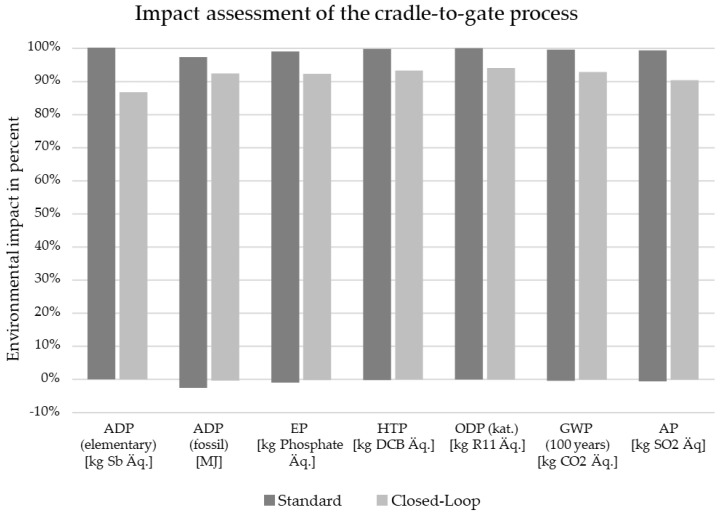
Environmental impacts, comparing life cycle assessment of the cradle-to-gate system boundary of the standard disposal process to the closed-loop recycling process investigated within the scope of this study. In total, 1000 manufactured pieces served as the functional unit.

## Data Availability

Not applicable.

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
