# Peer review of "Analysis of Mechanical Properties Related to Fiber Length of Closed-Loop-Recycled Offcuts of a Thermoplastic Fiber Composites (Organo Sheets)"

_materials, 2022, doi:10.3390/ma15113872_

Round 1

Reviewer 1 Report

It is a great honor to review this manuscript. The author's work is interesting, but I would like the authors to respond to the comments below.

  1. The abstract section should give more specific experimental data.
  2. There are too many keywords.
  3. Figure 3 first appears on line 148, while Figure 2 first appears on line 168. Therefore the order of Figures 2 and 3 should be reversed.
  4. The authors should give a clearer explanation of the reasons for the choice of key parameters, eg in a tabular format. It difficult for readers to extract key information from a large amount of text.
  5. In section 3.1, can you give pictures of the samples before and after the test?
  6. Section 3.5 seems out of the scope of the journal. The authors should pay more attention to the properties of the materials.
  7. The content of fibers and the orientation of fibers have a great influence on the properties of composite materials. Did the authors take this into account when preparing the samples?
  8. The glass fiber should be an inorganic non-metallic material. The author has always emphasized the recycling of organic materials. So the main purpose of the manuscript is to recycle glass fibers or organic matrices?
  9. Figure 2 shows that the sizes of the two flakes are quite different.

Reviewer 2 Report

This paper investigates the influences of recycled glass fiber-reinforced polypropylene on the mechanical properties of PP composites and its processability in the injection molding process. The manuscript is composed of an interesting introduction, followed by a detailed description of the preparation and investigation methods, which are coherently supported by the reported data. The authors conclude that the fiber lengths and fiber content are important factor affecting the mechanical properties of the final composites. Life cycle and profitability assessment are analyzed in order to verify the ecological and economic impact of the recycling approach. It is concluded that the closed-loop recycling of the offcuts is profitable from an ecological and economic point of view, and the mechanical properties are also sufficient.

The paper is well organized and the results are interesting and suitable for publishing on Materials. However, in order to improve the manuscript, I strongly recommend to compare their results with the huge amount of existing references and to report their new insights in this field.

Rheological properties of the composites are also important to better characterize the processing behavior of the resulting composites.

Round 2

Reviewer 1 Report

The author has solved most of the problems. In addition, the clarity of the figures in the manuscript should be improved.

Author Response

Comment 1: The clarity of the figures in the manuscript should be improved.

Response 1: The format and the presentation of the figures has been revised and some additions have been made to the captions.

In addition, the english language has been revised again.

Reviewer 2 Report

The manuscript was revised according to my requests. It can be published now.

Author Response

The english language of the whole manuscript has been revised again.